# The “ON-OFF” Switching Response of Reactive Oxygen Species in Acute Normobaric Hypoxia: Preliminary Outcome

**DOI:** 10.3390/ijms24044012

**Published:** 2023-02-16

**Authors:** Simona Mrakic-Sposta, Maristella Gussoni, Mauro Marzorati, Simone Porcelli, Gerardo Bosco, Costantino Balestra, Michela Montorsi, Claudio Lafortuna, Alessandra Vezzoli

**Affiliations:** 1Institute of Clinical Physiology, National Research Council (IFC-CNR), 20162 Milan, Italy; 2Institute of Chemical Sciences and Technologies “G. Natta”, National Research Council (SCITEC-CNR), 20133 Milan, Italy; 3Institute of Biomedical Technologies, National Research Council (ITB-CNR), 20090 Milan, Italy; 4Department of Molecular Medicine, Università di Pavia, 27100 Pavia, Italy; 5Department of Biomedical Sciences, University of Padova, 35131 Padova, Italy; 6Environmental, Occupational, Aging (Integrative) Physiology Laboratory, Haute Ecole Bruxelles-Brabant (HE2B), 1090 Brussels, Belgium; 7Anatomical Research and Clinical Studies, Vrije Universiteit Brussels (VUB), 1090 Brussels, Belgium; 8Department of Human Sciences and Promotion of the Quality of Life, San Raffaele Roma Open University, 00166 Roma, Italy

**Keywords:** oxidative stress, normobaric hypoxia, simulate altitude, ROS, electron paramagnetic resonance

## Abstract

Exposure to acute normobaric hypoxia (NH) elicits reactive oxygen species (ROS) accumulation, whose production kinetics and oxidative damage were here investigated. Nine subjects were monitored while breathing an NH mixture (0.125 F_I_O_2_ in air, about 4100 m) and during recovery with room air. ROS production was assessed by Electron Paramagnetic Resonance in capillary blood. Total antioxidant capacity, lipid peroxidation (TBARS and 8-iso-PFG2α), protein oxidation (PC) and DNA oxidation (8-OH-dG) were measured in plasma and/or urine. The ROS production rate (μmol·min^−1^) was monitored (5, 15, 30, 60, 120, 240 and 300 min). A production peak (+50%) was reached at 4 h. The on-transient kinetics, exponentially fitted (t_1/2_ = 30 min *r*^2^ = 0.995), were ascribable to the low O_2_ tension transition and the mirror-like related SpO_2_ decrease: 15 min: −12%; 60 min: −18%. The exposure did not seem to affect the prooxidant/antioxidant balance. Significant increases in PC (+88%) and 8-OH-dG (+67%) at 4 h in TBARS (+33%) one hour after hypoxia offset were also observed. General malaise was described by most of the subjects. Under acute NH, ROS production and oxidative damage resulted in time and SpO_2_-dependent reversible phenomena. The experimental model could be suitable for evaluating the acclimatation level, a key element in the context of mountain rescues in relation to technical/medical workers who have not had enough time for acclimatization—as, for example, during helicopter flights.

## 1. Introduction

Exposure to acute hypoxia elicits several functional changes in the human body to cope with the decreased oxygen availability [1]. Whereas some of these adjustments (increased pulmonary ventilation and heart rate) are well characterized, others are still poorly understood. Excesses in the accumulation of reactive oxygen species (ROS), generated within the mitochondria, drive mechanisms contributing to injury—including the induction of oxidative damage [2]. A growing body of evidence indicates that hypoxia decreases both the activity and effectiveness of the antioxidant system, as well as causing increased ROS production with a consequent increase in oxidative damage [3] to lipids [4,5,6], proteins and the DNA [5,7] of cellular compartments [8,9,10]. Most studies on the effects of hypoxia have been carried out at high altitude [9,10,11,12,13,14,15,16]. However, under such conditions, several factors other than hypoxia could induce oxidative damage; UV radiation, intense physical activity and cold may in fact produce an imbalance between ROS generation and antioxidant protection [17,18]. Indeed, a limited number of studies have been carried out in humans aimed at dissociating hypoxia from other potential environmental oxidative stressors and utilizing normobaric hypoxic gas mixtures in normobaric conditions [19,20,21]. Normobaric hypoxia (NH; 9 h simulating exposure at approximately 3800 m) was found to stimulate the production of cerebral ROS as well as associated biomarkers of oxidative stress [22]. However, to the best of our knowledge, the time course of changes in systemic ROS production, which represents the most valid tool for systemic oxidative stress evaluation during exposure to acute NH, has not still been assessed. 

In recent years, our group has developed an Electron Paramagnetic Resonance (EPR) microinvasive method [23,24,25,26,27,28,29,30] capable of providing absolute ROS quantification using a drop of capillary blood taken from the fingertip. The small amount of blood requested for the analysis and the ease of the sample collection procedure makes this technique particularly suitable for performing repeated measures in a short time frame and thus for quantitatively monitoring the ROS response kinetics (on and off) to a particular stimulus. The aim of this study was to investigate the time-course changes in systemic ROS production in response to acute, severe and short-term normobaric hypoxia (NH) exposure. The temporal relationship between ROS and oxidative stress biomarkers production was assessed in parallel. The present study is a continuation of the research project started in 2012 [31] that was continued with a field study in the Alps [13,14].

## 2. Results

### 2.1. ROS Production and SpO_2_ Kinetics

One subject dropped out during the experiment, after exactly 2 h of exposure to NH. The subject reported severe headache with dizziness and nausea. The other subjects completed the experimental session.

Exposure to acute NH induced EPR-detectable changes in the ROS production rate (μmol·min^−1^). ROS elicited by NH were measured at the different time points (5, 15, 30, 60, 120, 240 and 300 min) in subjects’ capillary blood. The data calculated at the different time points in the protocol (before, during and after NH exposure) are shown in Figure 1A. A production peak (+50%) was reached at 4 h of hypoxia exposure, followed by a return to around the pre-hypoxia level during the recovery phase while breathing room air. The exponential fit of the percent data of the on-transient kinetics returned a t_1/2_ = 30 min (*r*^2^
*=* 0.995; see insert at the right panel, bottom). 

No significant differences in ROS production or SpO_2_ were observed in the CTR group (Figure 1A, B) throughout the observational period. Compared to normoxia (0 min), acute NH induced a fast initial ROS production increase (15 min: +20%; 60 min: +36%) whose size appeared to be mirror-like in relation to the subjects’ blood oxygen saturation decrease—15 min: −12%; 60 min: −18%; see Figure 1B). The SpO_2_ (%) remained at a relatively constant level (85–90%) during the time course of the simulated hypoxia exposure. The relationship between the ROS production rate levels and the SpO_2_ (%) resulted in a significant, mild, inverse linear correlation (r = 0.43, *p* = 0.0001), obtained by the Spearman product–moment correlation (see Figure 1C).

### 2.2. Oxidative Damage and Antioxidant Capacity Response

Average TAC values did not show any significant change (Figure 2A). Plasmatic PC concentration (nmol·mg^−1^ protein) significantly (*p* < 0.05) increased after four hours of hypoxia exposure (at 240 min: +88%), whereas TBARS (μM) significantly (*p* < 0.05) increased (+33%) one hour after hypoxia offset. Both variables returned back to pre-hypoxia values during recovery (Figure 2B, C). After four hours of hypoxia exposure, the urinary 8-OH-dG concentration, a biomarker of DNA damage, was significantly increased (*p* < 0.05; +67%; Figure 2D); otherwise, no changes in urinary 8-iso-PGF2α, a marker of lipid peroxidation, were observed (Figure 2E). 

As shown in Figure 3, after 4 h of NH, subjects exposed to Hypoxia (F_I_O_2_ = 0.125) described a worsening of their feeling of overall wellness. With respect to the normoxia (black bars), a significant worsening in wellness and increase in tiredness and sleepiness—with no significant difference in the level of headache and nausea—were reported. In the CTR, only a little general malaise was reported—probably due to the 4 h spent sitting on the chair without the possibility of standing up. However, the items recorded in the VAS returned to the baseline within about a couple of hours.

## 3. Discussion

The present study was designed to analyze the kinetics of ROS production and oxidative damage in response to acute, severe and short-term normobaric hypoxia exposure. Nowadays, a number of reports have provided evidence for ROS production during acute hypoxia exposure, but a lack of information was found regarding its timing/on–off kinetics under a controlled environment, with the exclusion of the presence of external overlapping factors that can lead to confusion. 

The main finding of the present study was the significant and fast increase (+17%) in ROS production that occurred when breathing the hypoxic mixture (0.125% O_2_ in air). By the exponential fit of the transient on-kinetics, a t_1/2_ of 30.02 min (r^2^ = 0.995) was calculated. This was possibly ascribable to a transition to a low intracellular O_2_ tension, as evidenced by the significant reduction in SpO_2_. ROS production continued to increase, reaching a “plateau value” at 60 min that was maintained until the “turning off” of the hypoxia status. Thereafter, the ROS production returned to around normoxia levels (+9% with respect to the basal level). Nevertheless, a short (4 h) NH exposure did not seem to affect the prooxidant/antioxidant balance; no significant differences were found in the total antioxidant capacity (Figure 2A). By contrast, the decrease in the TAC values previously reported in response to hypoxia [9,15,16] could be due both to a longer exposure (>24 h) and/or to hypobaric conditions.

As also highlighted by Millet et al. (2012) [32], normobaric hypoxia can induce responses different from hypobaric hypoxia (HH); in fact, the physiological/biological responses to HH suggest it is a more severe environmental condition, leading to different physiological adaptations [33].

The total antioxidant capacity data did not evidence substantial changes during the NH experimental session (see Figure 2A), suggesting good tissue and systemic responses to the hypoxic stimulus; this is in agreement with Ribon and colleagues’ [33] data, who found that GPX and SOD activity was not significantly higher after NH. On the contrary, significant changes were observed after HH. 

The findings presented here were consistent with the fast increase of ROS concentrations found in isolated systemic vessels exposed to hypoxia [34,35,36]. The increase in ROS production in response to hypoxia has important systemic implications [37]—particularly at the brain level, with damage to the vascular endothelium, neurons and glia and the down-regulation of Na^+^-K^+^-ATPase, Ca^2+^-ATPase and Na^+^/Ca^2+^ exchanger activity [22,38]; these results have been recently confirmed by cellular microparticle production during acute hypoxia [39]. 

A possible explanation for the kinetics of this ROS production may be found in the autoxidation process of hemoglobin. As is well-known [40], hemoglobin continuously undergoes autoxidation by producing superoxide. In particular, a reduction in SpO_2_ has been found to result in a dramatic increase in the rate of hemoglobin autoxidation [40,41,42]. Therefore, because of the hypoxic exposition, oxidative stress might result from an increase in the autoxidation rate of partially oxygenated hemoglobin—particularly to that formed at the microcirculation level, as shown by the post-occlusion hyperhaemia reaction during acute hypoxia exposure [43]. This hypothesis seems to be confirmed by the reported significant inverse relationship between SpO_2_ and the ROS production rate presently detected. 

Furthermore, no changes in ROS production and SaO_2_ were observed in the CTR condition during the 4 h of the experiment, confirming that the data obtained from the EXP were not affected by variables (experiment duration, food restriction, tedium) other than the hypoxic status. Incidentally, these data also confirmed the high reproducibility of the EPR measurements and the stability of ROS production by biological samples in a full-rest context [23]. Indeed, the importance of an accurate and specific detection of reactive oxygen species in different biological samples is essential to the study of redox-regulated signaling in biological settings [44].

NH exposure induced an enhancement of oxidative damage, estimated by an increase in TBARS, PC and 8-OH-dG, that resulted in delays with respect to the ROS production increase. The late onset of oxidative stress damage might suggest to us that the oxidative damage phenomenon, evidenced by the production of some biomarkers, is a process growing in evidence over time. At the end of the NH exposure, in both markers of lipid peroxidation (TBARS in plasma and 8-iso-PGF2a in urine), no significant difference was found (see Figure 3C,E). One hour after the end of the experiment, the TBARS plasma concentration was found to be significantly (+33%) increased, returning to the pre-hypoxia control value in subsequent hours (Figure 2B). A similar behavior was also observed for PC plasma levels, which significantly increased during NH, attained their highest value (+88%) at 4 h, and then slowly decreased at hypoxia offset (Figure 2C). Finally, DNA underwent a 67% increase at 2 h of recovery from hypoxia (Figure 2D). 

VAS has been demonstrated to be a suitable method in the field of medicine to measure pain [45], general wellness [37,46], nausea [47], feelings of fatigue [48] and sleep quality [49]. Nevertheless, VAS has been also used as an auxiliary diagnostic method for the Lake Louise score (LLS) for evaluating acute mountain sickness (AMS) [50,51,52]. In hypobaric chambers, VAS shows the changing severity of symptoms during the process of elevation increasing [50,52]. However, from the results of the present study, a general discomfort, tiredness, headache and sleepiness during the time of the experiments were reported during the EXP, while only a general discomfort was reported in the CTR. As already pointed out, this could probably be ascribable to the subjects being compelled to sit on a chair with their mask attached to the instrument, with the impossibility of moving. This discomfort could perhaps be avoided by other experimental setups—for example, setting the experiment up in an extreme environments simulator center (i.e., NOI Techpark in Bolzano, South Tyrol, Italy), where climate chambers differ in size and equipment and can accommodate people for prolonged periods.

### Limitations

The present study had some experimental limitations, mainly concerning the number of participants, the duration of the experimental session (from four to six hours) and the investigated elevation, which was about 4100 m in altitude. The main purpose of the presently adopted protocol was to avoid any external and/or overlapping factors that can lead to confusion and in any way affect the results, such as those related to a prolonged permanence at high altitude. Indeed, the experimental design developed in the present study could be applied to a greater number of subjects, possibly of different age and/or levels of sport activity, and could increase both the amount and the duration of hypoxia. 

Finally, colorimetric assays are common methods for evaluating plasma TAC and TBARS, but their principal limitation is the non-specificity of the test. Nevertheless, various kits based on these methods are commercially available, and results have also been published recently [53,54,55]; they represent a reasonable compromise in terms of costs and reliability.

## 4. Materials and Methods

### 4.1. Subjects

Nine healthy sedentary subjects (7 males and 2 females; age 26.8 ± 13.2 years; height 175.2 ± 7.26 cm; weight 76.9 ± 10.1 kg) participated in this study (EXP); only 6 of these subjects (6 males; age 25.3 ± 1.5 years; height 1.74 ± 0.08 m; weight 75.43 ± 6.0 kg) lent themselves to the control condition (CTR). 

One week before the experimental session, all subjects underwent a clinical screening during which a physical examination and resting ECG were performed.

The exclusion criteria were regular smoking, hypertension, hypercholesterolemia, diabetes, cardiovascular or respiratory diseases, infections, supplementation of antioxidant or anti-inflammatory substances and habitual use of drugs.

All subjects resided below 500 m and, to minimize confounding effects, no subject had spent time above 1000 m of altitude in the four weeks preceding the study, nor was regularly engaged in a training program. Furthermore, participants were required to abstain from any strenuous physical activity and from alcohol and caffeine containing beverages, respectively, 2 weeks and 24 h prior to the study.

All subjects were informed about the aims, the experimental protocol and the potential risks of the investigation before giving written informed consent to take part in the study. This study was approved by the Local Ethical Committee (BESTA/IBFM, Report #27, 9 March 2016) and was carried out in accordance with the standard set by the Declaration of Helsinki [56].

### 4.2. Experimental Protocol

Participants arrived at the laboratory in Milan (122 m) early in the morning after a light breakfast and sat comfortably on a chair, breathing room air. The room temperature was kept constant at 21 °C. At time 0, the subjects in the EXP condition started breathing a normobaric hypoxic mixture (0.125 F_I_O_2_ in air, simulating about 4100 m altitude) obtained by removing oxygen from the air (MAG-10, Higher Peak LLC, Winchester, MA, USA). The gas mixture was delivered through a facemask at 30 L.min^−1^. Excess airflow was diverted outside the mask to prevent inspired oxygen pressure from increasing above 90 Torr. Four hours later, subjects were switched to normoxic breathing conditions. During the entire session, pulse oxygen saturation (SpO_2_, %) was monitored at the earlobe by an oximeter (Biox 3740 Pulse Oximeter, Ohmeda, Denver, CO, USA). Visual Analog Scale (VAS) scores were obtained both before the test session and at the end of the NH exposition to evaluate subjective general conditions. 

The subjects that lent themselves to the CTR condition were tested for the same time duration in room air (0.21 F_I_O_2_).

### 4.3. Sample Collection

With concern to the hypoxic condition (EXP), capillary blood samples for ROS assessments were collected before the test session while the participants were breathing room air, during hypoxia exposure (at 5, 15, 30, 60, 120, 180 and 240 min) and during the subsequent recovery phase with room air (at 15, 30 and 45 min). 

Venous blood samples for oxidative insult assessment (a cannula was placed into an antecubital vein) were withdrawn before the testing session while breathing room air, at different times during hypoxia (at 120 and 240 min) and during the recovery phase breathing room air (at 60 and 120 min). Blood samples (5 mL) were drawn and collected in heparinized tubes (Becton Dickinson Company, Swindon, UK) and centrifuged (Centrifuge 5702 R, Eppendorf, Hamburg, Germany) at 4000 rpm for 10 min at 4 °C. The separated plasma was stored in multiple aliquots at −80 °C. 

Urine samples were collected before and after hypoxia exposure (4 h) by voluntary voiding into a sterile container and were stored at −20 °C until assayed. All samples were thawed only once before analysis. In the normoxic condition (CTR), for each subject, capillary blood was drawn from the fingertip before and during 4 h of breathing (room air and normoxia at 30, 60, 120, 180, 240 and 285 min) under the same conditions as the EXP. The time points for the blood and urine sampling are shown in Figure 4**.**

### 4.4. Measurements

#### 4.4.1. ROS Detection by Electron Paramagnetic Resonance

For both the experimental procedures, ROS assessments were carried out by a X-band EPR Spectrometer (e-scan, Bruker, Bremen, Germany) and a 1-hydroxy-3-methoxycarbonyl-2,2,5,5tetramethyl-pyrrolidine-hydrochloride (CMH, Noxygen Science Transfer & Diagnostics, Elzach, Germany) spin-probe at 1 mM was prepared in buffer solution—Krebs-Hepes buffer (KHB) containing 25 μM deferroxamine methane-sulfonate salt (DF) chelating agent and 5 μM sodium diethyldithio-carbamate trihydrate (DETC) at pH 7.4—which reacts with extra and intracellular one-electron oxidants, generating the EPR detectable radical CM•. Capillary blood was immediately treated with CMH (1:1) and put into an EPR capillary tube and then placed inside the cavity of the EPR spectrometer for acquisition at 37 °C by the temperature and Gas controller “Bio III” unit (Noxygen Science Transfer & Diagnostics GmbH, Germany), interfaced to the spectrometer. 

All spectra were collected by adopting the same acquisition parameters: microwave frequency = 9.652 GHz; modulation frequency: 86 kHz; modulation amplitude: 2.28 G; center field: 3456.8 G; sweep width: 60 G; microwave power: 21.90 mW; number of scans: 10; receiver gain: 3.17 × 10. Data were converted in absolute concentration values (μmol·min^−1^) by using the CP• (3-carboxy-2,2,5,5-tetramethyl-1-pyrrolidinyloxy) stable radical as an external reference. Spectra acquired were recorded and analyzed using the Win EPR software (version 2.11) standardly supplied by Bruker. 

Details on the procedures have been previously reported [9,10,23,24]. 

#### 4.4.2. Total Antioxidant Capacity (TAC)

TAC was measured using a commercial enzymatic kit (Cayman Chemical, Ann Arbor, MI, USA). The assay relies on the ability of antioxidants present in plasma to inhibit the oxidation of ABTS (2,2′-Azino-di-[3-ethylbenzthiazoline sulphonate]). The TAC signal is proportional to the suppression of the oxidized ABTS absorbance signal at 750 nm and is evaluated by a Trolox standard curve [24,57].

#### 4.4.3. Protein Carbonyls (PCs)

The accumulation of oxidized proteins was measured by their reactive carbonyl content. A protein carbonyl (PC) assay kit (Cayman Chemical, Ann Arbor, MI, USA) was used to colorimetrically evaluate the oxidized proteins at 370 nm. The obtained values were normalized to the total protein concentration (at 280 nm) to determine the protein loss during the washing steps, as suggested by the kit’s user manual [9,10,23,24]. 

#### 4.4.4. Thiobarbituric Acid-Reactive Substances (TBARS)

TBARS measurements were adopted to detect lipid peroxidation. An assay kit (Cayman Chemical, Ann Arbor, MI, USA) was used to allow a rapid photometric thiobarbituric acid malondialdehyde (TBAMDA) adduct detection at 532 nm. A linear calibration curve was computed from pure malondialdehyde-containing solutions [9,10,16,23,24,57,58].

#### 4.4.5. 8-Isoprostane (8-iso-PGF2α)

Lipid peroxidation was assessed also using an immunoassay of 8-isoprostane concentration (Cayman Chemical, Ann Arbor, MI, USA) in urine, as previously described [9,10,14]. Samples and standard were read in duplicate at a wavelength of 512 nm. Results were normalized by urine creatinine values.

#### 4.4.6. 8-OH-2-Deoxyguanosine (8-OH-dG)

Levels of 8-OH-dG were measured using an immunoassay EIA kit (Cayman Chemical, Ann Arbor, MI, USA) in urine. This is a biomarker for DNA damage assessment. Samples and standards were spectrophotometrically read at 412 nm. Results were normalized by urine creatinine values [10,29,59].

#### 4.4.7. Visual Analog Scale (VAS)

Subjective mood, general wellness/malaise, rested/tired feelings, headaches, sleepiness and nausea were evaluated using a 0–100 mm visual analog scale (VAS). This scoring system has been previously suggested for assessing discomfort and/or general malaise [52]. Based on its usefulness in performing other clinical evaluations, VAS was deemed suitable for testing the subjective perception of normobaric hypoxia effects [20,37,60].

### 4.5. Statistical Analysis

All values are expressed as mean ± SD. Data were analyzed using parametric statistics following the mathematical confirmation of a normal distribution using the Shapiro–Wilks *W* test. To determine the statistical significance of the changes during hypoxia exposure, ANOVA for repeated measures was performed, followed by the Tukey’s post-hoc test. Significant differences were set at *p* < 0.05. The relationships between the investigated variables were assessed using Spearman correlation coefficients. Change Δ% estimation (([post value-pre value]/pre value) × 100)) was also reported in the text. Statistical analyses were performed using Prism 9.3.1 software for Mac (GraphPad, Software Inc., San Diego, CA, USA).

ROS production was considered as the primary outcome (no other parameters were taken into account) and prospective calculations of power to determine sample size were made using G power software (GPower 3.1) [61]. At 80% power, the sample size—calculated in preliminary studies [9,29]—was set at five subjects.

## 5. Conclusions

In conclusion, the results of the present study provide evidence that under short, acute, normobaric hypoxia, ROS production and oxidative damage are time as well as SpO_2_-dependent and are reversible phenomena. At the same time, the increase in oxidative damage, measured by enzymatic assays, appeared to be delayed with respect to the production of ROS—these latter EPR measurements being assessed by a mini-invasive absolute quantitative method. Lastly, the subjective evaluation of general physical conditions revealed a number of moderate-intensity discomforts described by most of the subjects, excluding nausea and headache. These findings seem overall to suggest that the experimental design proposed here can be considered a suitable model for use in future research studies. In fact, the altitude and time of exposure chosen tended to simulate the conditions experienced by mountain lovers during a single-day trip in alpine environments. Moreover, the study might help to highlight some key elements in acclimatization, as well as being of help for all those people who work at high altitudes without sufficient time to acclimatize, such as mountain rescue personnel/teams operating on helicopters during emergency medical operations (i.e., pilots, mountain rescuers and medical doctors) [43]. Finally, our EPR methods could be applied as a standard procedure in mountain medicine for studying changes induced by hypoxia due to acute and chronic high altitude stays.

## Figures and Tables

**Figure 1 ijms-24-04012-f001:**
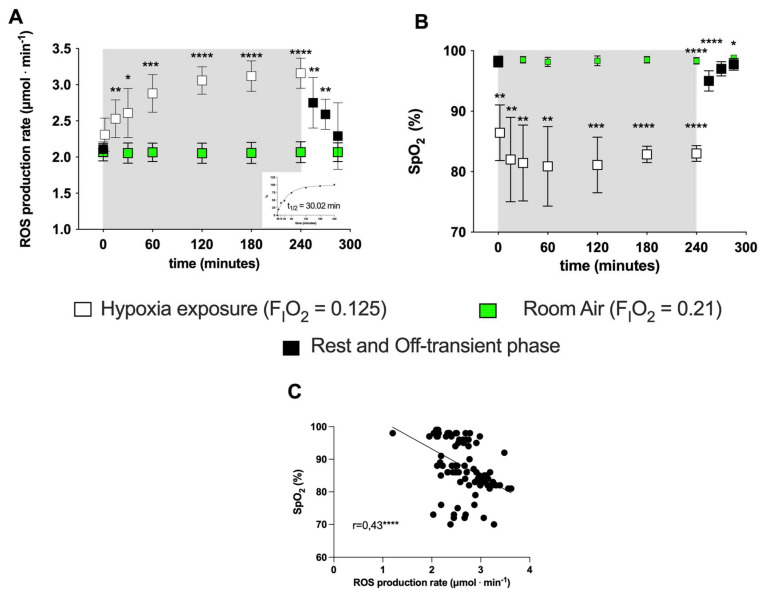
(**A**) Time course of the ROS production rate (μmol·min^−1^) assessed by EPR technique and (**B**) SpO_2_ (%) during the transition from normoxia to acute normobaric hypoxia and back to normoxia during recovery. EXP group: full symbols are the data obtained before (0 min) and 15, 30, 45 min after hypoxia exposure (recovery); open symbols are the data during hypoxia status: at 5, 15, 30, 60, 120, 180, 240 min. Green symbols are the data obtained from the CTR (room air, normoxia state FiO_2_ = 0.21) for ROS (μmol·min^−1^) and SpO_2_ (%). In the inset at the right bottom of Figure A, the exponential fit of the on-transient kinetics data (t_1/2_ = 30.02 min, r^2^ = 0.995) is shown. Results are expressed as mean ± SD. (**C**) Data (full symbols) and linear correlation line (continuous line, r = 0.43) of the ROS production rate versus SpO_2_. Changes over time were significantly different when compared to pre-exposure levels (* *p* < 0.05; ** *p* < 0.01; *** *p* < 0.001; **** *p* < 0.0001 symbols).

**Figure 2 ijms-24-04012-f002:**
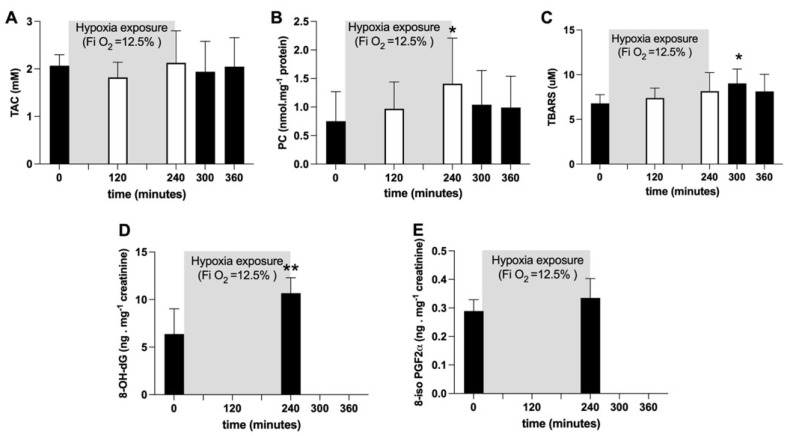
Histogram plots (mean ± SD) of (**A**) TAC (mM), (**B**) TBARS (µM) and (**C**) PC (nmol·mg^−1^ protein) concentration levels obtained from plasma samples collected at 0, 120 and 240 min of NH. Data at 60 and 120 min of the recovery time, when breathing normoxic room air, are also shown. (**D**) 8-OH-dG and (**E**) 8-iso-PGF2α (ng·mg^−1^ creatinine) data levels before and after NH. Changes over time were significantly different when compared to pre-exposure levels (* *p* < 0.05; ** *p* < 0.01 symbols).

**Figure 3 ijms-24-04012-f003:**
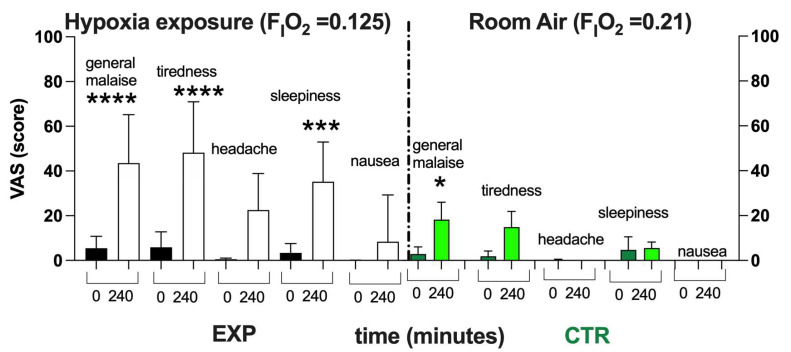
Histogram plot (mean ± SD) of the VAS score from the EXP and CTR. Data obtained Pre- (black and green bars respectively) and after 240 min of NH (white and light green bars, respectively) are shown. Changes over time were significantly different when compared to pre-exposure levels (* *p* < 0.05; *** *p* < 0.001; **** *p* < 0.0001 symbols).

**Figure 4 ijms-24-04012-f004:**
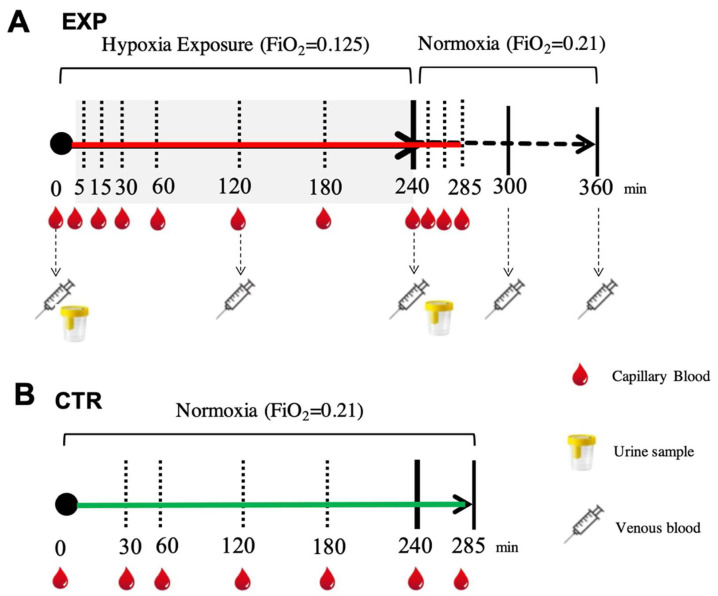
(**A**) EXP. Experimental timeline adopted for collecting each subject’s blood sample (capillary (red drops), venous (syringes)) to measure ROS production by EPR, oxidative damage (PC and TBARS) by enzymatic assays and urine (glasses) to assess urinary 8-OH-dG and 8-iso-PGF2 alfa by ELISA. (**B**) CTR. Experimental timeline (continuous green line) adopted for collecting capillary blood samples (red drops) to measure ROS production by EPR.

## Data Availability

Data are available at request from the authors.

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
