# Peer review of "The “ON-OFF” Switching Response of Reactive Oxygen Species in Acute Normobaric Hypoxia: Preliminary Outcome"

_ijms, 2023, doi:10.3390/ijms24044012_

Round 1

Reviewer 1 Report

The current study aimed to investigate changes in redox homeostasis during short-term normobaric hypoxia.

In the study are interesting aspects, but the data presented here are far too preliminary and solid conclusions cannot be reached.

-         - Apart from only 9 subjects, the data would require a control group of healthy people giving blood and urine at the same time as the experimental group to exclude changes due to the experimental design, especially since the authors also found changes in well-being, tiredness and sleepiness of the control group.

-     -  Furthermore, the groups could also be subdivided to compare the data between trained and untrained persons, since previous studies suggest that moderate physical activity may attenuate altitude-induced oxidative stress at least during long-term hypoxic stress

- The variation between the experimental and the control group is too strong, e. g. the total number, different genders and the age.

Author Response

Comments and Suggestions for Authors

The current study aimed to investigate changes in redox homeostasis during short-term normobaric hypoxia.

In the study are interesting aspects, but the data presented here are far too preliminary and solid conclusions cannot be reached. 

-Apart from only 9 subjects, the data would require a control group of healthy people giving blood and urine at the same time as the experimental group to exclude changes due to the experimental design, especially since the authors also found changes in well-being, tiredness and sleepiness of the control group. 

The authors thank this reviewer for his/her positive comments. We agree about the small number of subjects (n=9): aware of it, we specifically pointed this out among the experimental limitations in the Limitations section of the previously submitted manuscript. However, we would like to underline that these subjects underwent two experimental sessions: one in normobaric hypoxia, the other in the air.  This experimental design was planned to compare the same subjects and avoid handling two different groups, which could lead us to confusion in the interpretation of the results. This item has now been added in text.

Finally, tiredness and sleepiness, as shown in figure 3, are present in both cases (EXP and CTR), even if in the CTR (room air) without statistically significant evidence. The only significant symptom in the CTR condition was a general malaise, due to the boredom of sitting for 4 hours doing nothing. Finally, for better clarity, “preliminary outcome” was added to the Title.

-     -  Furthermore, the groups could also be subdivided to compare the data between trained and untrained persons, since previous studies suggest that moderate physical activity may attenuate altitude-induced oxidative stress at least during long-term hypoxic stress

All subjects participating to the study were sedentary, as already reported in the manuscript (see section 4.1)

- The variation between the experimental and the control group is too strong, e. g. the total number, different genders and the age.

The authors agree with the reviewer and thank him/her for the remark. However, the variation found between experimental and control groups depended on the fact that the two females and the oldest male subjects did not underwent to the CTR session. This relevant observation is reported in the revised manuscript.

Reviewer 2 Report

Major comments:

In section 4.4.1: For Electron paramagnetic resonance spectrometry, the authors should write in detail the acquisition parameters.

Also, did the authors perform the simulation of the spectra obtained? If yes, it is important to present it either in the manuscript and/or the supplementary dataset.

It is very inconvenient for a reviewer or a reader to go through the results before explaining the materials used and the methods employed to achieve the results. Therefore I suggest making the section on the materials and methods before the results and discussion.

Formatting errors

Line 49: delete additional bracket

Line 95, symbol, same in line 100

Please check for minor errors/ typos throughout the text.

Author Response

Comments and Suggestions for Authors

Major comments:

In section 4.4.1: For Electron paramagnetic resonance spectrometry, the authors should write in detail the acquisition parameters.

The authors thank this reviewer for the remark, the EPR acquisition parameters were added in the revised version of the manuscript (see Section 4.4.1.)

Also, did the authors perform the simulation of the spectra obtained? If yes, it is important to present it either in the manuscript and/or the supplementary dataset.

The EPR signal of the spectra analyzed in this study is a triplet coming from the interaction of the 14N–OH group of the probe CMH with the ROS oxygen unpaired electron (NOH + O2 → NO + H2O2). In other words, the silent CMH molecule when encountering ROS oxygen becomes in turn paramagnetic so giving the triplet EPR signal. On the other hand, the equivalent protons do not give rise to hyperfine coupling splitting so that a simple spectrum is recorded and the use of a Simulation Software, powerful tool for spectra interpretation, is unnecessary. Calibration curves and test experiments confirming the high reproducibility of the EPR measurements were instead previously reported by some of us. In particular, the EPR signal is proportional to the unpaired electron numbers and can, in turn, be transformed in absolute produced micromoles (μmol · min−1): the stable CP (3-Carboxy-2,2,5,5-tetramethyl-1-pyrrolidinyloxy) radical signal was used as external reference for quantitative determination. Examples of stacked plots of the ROS EPR spectra are previously published in: Mrakic-Sposta et al, 2012; 2014; 2015a,b; 2018;  2022; Bosco et al, 2022.

It is very inconvenient for a reviewer or a reader to go through the results before explaining the materials used and the methods employed to achieve the results. Therefore I suggest making the section on the materials and methods before the results and discussion

The authors thank this reviewer for his/her remark and might agree. However, the format required by the editorial office of MDPI for IJMS, of course independent of the authors, was followed.

Formatting errors

Line 49: delete additional bracket

Thanks, done

Line 95, symbol, same in line 100

Thanks, done

Please check for minor errors/ typos throughout the text.

Thanks, done

Reviewer 3 Report

This is a rather straightforward study by experienced groups that address oxidative stress in acute hypobaric hypoxia (HH) in-vivo experimental study at high altitude (9-16). However, under such conditions several environmental oxidative stressors other than simple hypoxia could induce oxidative damage. In this manuscript, the authors investigated the time-course changes of the systemic ROS production in response to in acute normobaric hypoxia (NH) exposure using their developed Electron Paramagnetic Resonance (EPR) microinvasive method (23-30) in nine healthy sedentary subjects.

They found that a ROS production peak (+ 50%, t1/2 = 30.02 min) was reached at 4 hours of hypoxia exposure followed by a return to about the pre-hypoxia level during the recovery phase, when breathing room air (figure 1A) with the mirror-like related SpO2 decrease (figure 1B). The relationship between the ROS production rate levels and the SpO2 (%) resulted in a significant mild inverse linear correlation (r=0.43, p=0.0001) (figure 1C). Some of the oxidative damage markers temporally increased at 4 hours of hypoxia exposure (figure 2). Subjects exposed to hypoxia (FIO2=0,125) referred a worsening of their feeling of overall wellness excluding nausea and headache (figure 3).

           Based on the above results they propose that the experimental design here can be considered as a suitable model to evaluate the acclimatation level, key element in a context of mountain rescue, in relation to technical/medical workers who had no time enough for acclimatization, as for example during helicopter flights.

Although the small number of participants and the duration of the experimental session from 4 to 6 hours, this work is a nice contribution to the field of human physiological response in acute normobaric hypoxia (NH).

I have few concerns to address as below.

Major point:

Is the increased Visual Analog Scale (VAS) from EXP returned to the level from CNT group during the recovery phase in figure 3?

Minor point:

Figure 2A and Bshould be Figure 1A and B (lane 86).

Author Response

The authors thank this reviewer for his/her positive comments.

I have few concerns to address as below.

Major point:

Is the increased Visual Analog Scale (VAS) from EXP returned to the level from CNT group during the recovery phase in figure 3?

Thanks for the remark. Yes, even if not shown in the Figure, all the items recorded in the VAS returned to the baseline within a couple of hours. This point was now clarified in the revised manuscript (see results)

Minor point:

Figure 2A and B should be Figure 1A and B (lane 86).

The authors thank the reviewer for the remark. The number of the Figures is now rightly reported.

Round 2

Reviewer 1 Report

The authors added "preliminary results" to the title, clearly indicating that further studies are needed. Thus, my concerns of the previous review are addressed.

Reviewer 2 Report

The authors have addressed my points. It can be accepted in the current form.